# The Comparison between Herniated and Non-Herniated Disc Levels Regarding Intervertebral Disc Space Height and Disc Degeneration, A Magnetic Resonance Study

**DOI:** 10.3390/diagnostics13203190

**Published:** 2023-10-12

**Authors:** Türkhun Çetin, Sevket Kahraman, Volkan Kızılgöz, Sonay Aydın

**Affiliations:** Department of Radiology, Faculty of Medicine, Erzincan Binali Yıldırım University, 24100 Erzincan, Turkey; sevketkahraman92@gmail.com (S.K.); volkankizilgoz@gmail.com (V.K.); sonay.aydin@erzincan.edu.tr (S.A.)

**Keywords:** lumbar vertebrae, Pfirrmann grading system, intervertebral disc degeneration, intervertebral disc displacement, magnetic resonance imaging

## Abstract

Purpose: The main purpose of this study was to evaluate the intervertebral disc height and intervertebral disc degeneration between the normal group and the group with disc herniation at the level of the L4–L5 intervertebral disc by MRI using the Pfirrmann grading system. Materials and Methods: 385 patients were included in this study. MRI images were reevaluated and intervertebral disc heights were measured from the anterior, middle and posterior segments. Researchers divided disc pathologies into two groups. In the non-herniated group; normal or bulging ones; in the herniated group, they included those with protrusion or extrusion. Results: 385 lumbar MRI examinations meeting the study criteria were included in study. There were 56.9% (219/385) females and 43.1% (166/385) males in the study. For the whole patients in the study group, the intervertebral disc height values at the L4–5 level were measured as 12.34 mm, 11.58 mm, and 7.60 mm in the anterior, middle, and posterior localizations, respectively. Conclusions: At the L4–5 level, the height of the disc distances in the herniated group was lower than in the normal group. The Pfirrman score was found to be higher in the herniated group in terms of disc degeneration compared to the normal group.

## 1. Introduction

One of the most common causes of low back pain and sciatic pain in adults is lumbar disc herniation (LDH) [1]. Lumbar disc herniation (LDH) is the rupture of the annulus fibrosus as a result of lumbar degeneration or prolonged strain. The nucleus pulposus travels laterally or backward, protruding behind the torn annulus or quite laterally [2,3]. The bulging disc’s compression of the dorsal and/or ventral nerve roots causes low back discomfort, leg pain (sciatica), muscular spasm and trunk restriction movement. Lumbar disc herniation affects an estimated 5–15% of individuals experiencing low back discomfort. Lumbar degenerative herniation (LDH) represents the prevailing spinal problem necessitating surgical intervention. According to clinical standards, it is recommended to do a comprehensive history taking and physical examination in order to effectively rule out the diagnosis of lumbar disc herniation (LDH). Nevertheless, the diagnostic precision of both the process of obtaining patient history and conducting physical examinations remains inadequate. The utilization of diagnostic imaging in individuals experiencing back pain and/or leg discomfort is frequently employed to evaluate the presence of nerve root compression resulting from disc herniation or spinal stenosis, as well as cauda equina syndrome. In addition, diagnostic imaging can be employed to ascertain the specific disc level that is impacted prior to surgical intervention.

Diagnostic imaging techniques often employed in medical practice include Magnetic Resonance Imaging (MRI), Computed Tomography (CT), X-ray, and myelography. At present, magnetic resonance imaging (MRI) is the preferred modality for medical imaging due to its inherent benefits of non-ionizing radiation and excellent visualization capabilities, particularly for soft tissue. Computed tomography (CT) is frequently employed and readily accessible for the identification of morphological alterations, and it is widely acknowledged for its diagnostic utility in herniated disc cases. In contrast to magnetic resonance imaging (MRI), computed tomography (CT) offers a more cost-effective alternative, shorter overall testing duration, and more accessibility inside hospital environments. However, it is important to note that CT scanning has the potential downside of exposing individuals to ionizing radiation. Myelography is a diagnostic procedure that entails the administration of a contrast material into the lumbar spine, subsequently followed by the acquisition of X-ray, CT, or MRI images. In specific circumstances, such as the presence of metal implants or spinal malalignment, myelography may be preferred over magnetic resonance imaging (MRI) as the primary imaging modality. Plain radiography, sometimes known as X-ray imaging, is widely utilized in medical practice primarily because of its affordability and widespread accessibility [1].

Lumbar disc degeneration was characterized as a decrease in intervertebral disc height and a radiological decrease in signal intensity on T2-weighted magnetic resonance imaging. The bulging of the intervertebral disk occurs when the nucleus pulposus experiences a decrease in turgor and the anulus loses its flexibility, resulting in the disk protruding beyond the boundaries of the vertebral body. The occurrence of herniation of the nucleus pulposus (HNP) is characterized by the protrusion of the disk material beyond the boundaries of the surrounding vertebral endplate due to an anular defect. This condition is commonly referred to as disk herniation. Trauma represents the predominant factor leading to the rupture of the nucleus pulposus via the anulus fibrosus. The outcome entails the protrusion or extrusion of the disk material into the spinal canal [4,5].

Radiologically, lumbar disc degeneration, lumbar disc herniation, and osteoporosis increase with age [6,7]. The parameters that are suggestive of disc degeneration, disc height and disc signal intensity changes were reported in previous studies [4,8,9,10,11,12]. To define lumbar degenerative disc disease, two classification schemes are often employed. The Modic classification (MOD) scheme [13,14], which includes contiguous vertebra end plate alterations, is one of them. On the other hand, the Pfirmann classification system [14,15] considers imaging qualities, signal intensity, and disc height. MRI is useful for diagnosing lumbar disc disorders and herniations [4,16].

The aim of our study was to reveal the intervertebral disc space height and disc degeneration level differences between herniated and non-herniated patients. The intervertebral disc height measurements and qualitative disc degeneration assessments were compared between herniated and non-herniated patients to find out any significant difference between these groups. Lumbar disc herniations are mostly encountered at lower lumbar levels and the L4–5 level is one of the most effected intervertebral disc levels in the lumbar region. In this investigation, the discs of patients in the L4–5 level were sampled regarding the disc degeneration levels using the Pfirmann classification, and intervertebral disc heights were measured using T2-weighted sequences via MRI. The authors thought that it would be beneficial to reveal these data to be used in disc assessments, contribute to the literature and to be used in further studies.

## 2. Materials and Methods

This retrospective study was approved by our institutional review board. The ethics committee has waived the informed consent from each patient due to the study’s retrospective nature.

### 2.1. Study Population

The patients who were undergone lumbar MRI for various reasons including lumbar back pain, radiculopathy, central canal stenosis, physical examination consistent with lumbar disc herniation or degenerative diseases of the spine, suspicious symptoms for infection, risk of spinal metastasis, scoliosis or lordosis were re-interpreted regarding the disc degenerations and intervertebral disc height measurements. The data for this investigation were reassessed using the lumbar MRI scans of 789 individuals between 1 July 2022 and 30 December 2022. L4–5 intervertebral disc levels of all these patients were re-interpreted and two groups were created from the dataset for our investigation.

Normal lumbar MRI images and only those with pathology in L4–5 intervertebral disc were included in the study. In the non-herniated group, there was no disc pathology at any level of the lumbar intervertebral disc or there was bulging only in the L4–5 intervertebral disc. In the herniated group, only L4–5 intervertebral disc protrusion or extrusion was present. Herniated group had intervertebral disc protrusion or extrusion at this level. Patients under 18 years of age were excluded from the study to isolate the skeletally mature patients, which consists of the patient group to focus on this investigation.

Those patients who had MRIs with motion artifacts, under 18 years old in the herniated group, with a history of lumbar surgery, neoplasm, infection, fracture, scoliosis, or any congenital spinal pathologies, were excluded from the study. Patients with motion artifacts in magnetic resonance (MR) images (*n* = 40), under 18 years of age (*n* = 25) in the herniated group, with multi segmental disc pathology or with disc hernias in other lumbar levels (*n* = 264), a history of lumbar surgery or internal fixation material due to lumbar surgery (*n* = 32), vertebral neoplasm (*n* = 12), infectious conditions such as discitis (*n* = 12), vertebral fracture (*n* = 4), scoliosis (Cobb angle ≥ 20°) (*n* = 11), congenital pathologies (*n* = 4) (one patient with diastematomyelia, two patients with tethered cord, one patient with dural ectasia), a total of 404 patients were excluded from the study. Following the submission of exclusion criteria, 385 individuals (219 women, 166 men, mean age: 46.68 ± 16.17 years) were accepted into the study (Figure 1). In the herniated group, there were 100 patients (57 women, 43 males, mean age: 52.25 ± 14.21 years). The non-herniated group had 285 patients (162 women, 123 men, mean age: 44.73 ± 16.38 years).

### 2.2. Magnetic Resonance Imaging Protocol

MR Imaging was performed with a 1.5 T MRI (Magnetom Aera, Siemens, Erlangen, Germany) device using a (32 channel) lumbar coil. Patients were given a standard position to obtain a lumbar MRI, lying on his back. T1 sagittal, T2 sagittal and T2 axial images were obtained for lumbar imaging. Sagittal plane T2-weighted images (TR [Time of Repetation]: 4120 ms, TE [Time of Echo]: 104 ms, average: 2, field of view: 280 mm, slice thickness: 4 mm, voxel size: 0.9 × 0.9 × 4 mm), sagittal plane T1-weighted images (TR: 646 ms, TE: 9 ms, average: 2, field of view: 280 mm, slice thickness: 4 mm, voxel size: 0.9 × 0.9 × 4 mm), and axial plane T2-weighted images (TR: 5070 ms, TE: 88 ms, average: 1, field of view: 190 mm, slice thickness: 4 mm, voxel size: 0.7 × 0.7 × 4 mm) were obtained from each patient as a part of a routine lumbar spinal study with regard to intervertebral disc protocol. Two radiologists with 25 years and five years of experience re-evaluated and measured all images. Both reviewers evaluated the images and done the measurements with consensus, at the same time, using the same monitor. The MR images were analyzed, and measurements were made in standard digital imaging and medical formats using a picture archiving and communication system (Akgün PACS Viewer v7.5, Akgün Software, Ankara, Turkey).

### 2.3. Image Analysis and Measurements

At the L4–L5 intervertebral disc level, measurements with sagittal T2-weighted MRI sequence and Pfirmann grading [15] for disc degeneration were performed (Figure 2). Disc pathologies were classified with axial T2-weighted MRI sequence. The Pfirrmann grading system is divided into five categories. It is accepted as a reliable method for classifying disc degeneration in MRI scans utilizing parameters such as disc structure, height, signal qualities, and nucleus and annulus differentiation [15,17].

At the L4–L5 intervertebral disc level, the distances between the lower plateau of the L4 vertebral corpus and the upper plateau of the L5 vertebral corpus were measured anteriorly, medially, and posteriorly. Anterior and posterior disc space heights were measured 2 mm distant from the corners of corpus vertebrae. The middle intervertebral disc space height was measured from the midpoint of the total anteroposterior disc space diameter (Figure 3). If there were osteophytic protrusions from the vertebral corpus, and the anteroposterior diameter of inter vertebral disc space was elongated in shape, the average anteroposterior length of upper and lower vertebra articular plateaus was used. All height measurements were done from cortex to cortex and the measurement results were noted using two digits after comma as viewed by the monitor of our PACS system. All intervertebral disc space heights were measured twice and the average value was considered as the result to be used for statistical analysis.

The outside borders of the lumbar intervertebral discs were used to classify them as bulging, protruding, or extruding. The disc was considered bulging if more than one-quarter of its circumference was pushed beyond the vertebral body borders. If the outer borders of the herniated section of the intervertebral disc were smaller than the measured distance at the base of the herniated part of the disc, it was deemed protruding. If the distance measured between the edges of the herniated and nonherniated intervertebral discs was larger than the length at the base of the herniation in at least one plane of the MRI, it was termed an extrusion [18]. If the herniated disc material has lost its continuity with the main disc, it is called sequestration and is one of the subclasses of extrusion. In this study, bulging was not considered as a herniation. Individuals with normal discs or disc bulging were classified as the non-herniated group. Those with protrusion or extrusion were classified as the herniated group.

### 2.4. Statistical Analysis

Statistical analysis was performed using the SPSS version 22.0 software program for Windows. Descriptive statistics include frequency, percentage, mean, standard deviation, median, and min-max values. Non-parametric test procedures were used if the quantitative values did not reveal a normal data distribution in the Kolmogorov-Smirnov test. In this context, Mann Whitney U Test was used to determine the relationships between parameters. Chi-square test was used in the pairwise analysis of qualitative data. The results were evaluated within the %95 confidence interval, and the *p*-values less than 0.05 were considered statistically significant.

## 3. Results

Lumbar MRI examinations of 789 patients were scanned in this study. 385 lumbar MRI examinations meeting the study criteria were included in our study. There were 56.9% (219/385) females and 43.1% (166/385) males in the study (Table 1). The mean age in the study population was 46; in the control group was 44, and in the herniated group, it was 54 years. 74% (285/385) of the study population consisted of the control group (Table 2).

For the whole patients in the study group, the intervertebral disc height values at the L4–5 level were measured as 12.34 mm, 11.58 mm, and 7.60 mm in the anterior, middle, and posterior localizations, respectively (Table 3). In the control group, the intervertebral disc height values at the L4–5 level were measured as 12.57 mm, 11.92 mm, and 7.79 mm for the anterior, middle and posterior height measurements, respectively (Table 4). In the herniated group, the intervertebral disc heights were 11.69 mm, 10.61 mm, and 7.07 mm for the anterior, middle and posterior measurements, respectively (Table 5).

The distribution of the total of patients, control group and herniated group in L4–5 intervertebral disc according to Pfirrmann’s grading system are shown in Table 6, Table 7 and Table 8.

A statistically significant relationship was observed between the groups in terms of age. The mean age of the herniated group was higher than the normal group. There was no statistically significant relationship between the groups in terms of gender.

The mean height value of the intervertebral disc at the L4–5 level of the normal group was statistically significantly higher than the herniated group. At the L4–5 level, the intervertebral disc space’s anterior, middle and posterior height in the control group was significantly higher than in the herniated group.

There is a statistically significant relationship between the groups in terms of Pfirrmann grade. The herniated group’s Pfirrmann grade was significantly higher than the control group.

## 4. Discussion

Intervertebral discs connect the vertebral bodies and transmit the axial load [19]. The identification of disc herniation is a routine responsibility for radiologists. While the process of diagnosing a condition is typically uncomplicated, it can provide challenges in cases where there is a displacement of intervertebral discs. The potential oversight of disc herniations occurring in the neural foramen or in its lateral vicinity can be mitigated by acquiring sagittal images with sufficient lateral coverage. However, it is worth noting that these herniations are typically most effectively visualized on coronal scans. Preoperative identification of intradural disc herniations is crucial due to their characteristic morphology, as it enables the surgeon to anticipate the necessity of doing a durotomy and subsequent dural repair. Sequestered intervertebral discs frequently exhibit a lentiform shape and tend to be inconspicuous when observed on sagittal images. The potential for missing them exists when axial images are solely acquired at the intervertebral disc level. When evaluating magnetic resonance imaging tests, it is important to consider other conditions that may resemble disc herniation. These conditions encompass disc osteophyte complex, epidural hematoma, facet joint cyst, as well as tumors such meningioma, nerve sheath tumors, and metastases. Accurate anatomical knowledge of intervertebral discs is important for clinicians in diagnosis of radiculopathies or interventions such as stabilization of various spinal deformities [20]. Since the lumbosacral region is mobile, it is prone to disc herniation. As mentioned in previous studies, disc herniation occurs mostly (75%) in the L5-S1 lumbosacral region and then at the level of the L4–L5 intervertebral disc in 15–20% [21]. We used the L4–5 level to standardize the disc assessments for this study.

The overall height of the vertebral column is typically regarded to equal one-quarter of the sum of the disc heights [22]. In their studies, some researchers did not measure disc distances but instead used converted ratios and measurements based on disc heights [19]. Some studies examined disc height and end plate shape in order to understand its relationship with disc pathologies. According to the literature, spinal levels with concave-shaped end plates may have much greater discs than flat-shaped levels for deteriorated and herniated intervertebral discs. In degenerative discs, flat-shaped levels had considerably higher average disc height than levels with irregular-shaped end plates, but not in herniated discs [23]. In this current research, we did not focus on specific shapes of endplates; instead, we measured the intervertebral disc distance at the L4–5 level anteriorly, in the middle, and in the posterior. At this level, we contributed to the literature in terms of its relationship with the Pfirrmann grade and the disc space height measurements between the non-herniated and herniated groups. Several models for describing degenerative lumbar disc disease have been researched and produced. Pfirrmann classification is currently one of the most widely used, based on MRI analysis and four parameters: disc structure, the distinction between nucleus and annulus, disc intensity and height. The discs are grouped into five categories based on these parameters. Nonetheless, Pfirrmann’s classification [15] does not take angulation losses or anterior-posterior height discrepancies into account, leaving the sagittal balance of the spine unaffected [24]. Mirab et al. reported the anterior, middle, and posterior disc heights at the level of L4–5 intervertebral discs to be 18.14 mm, 13.82 mm, and 10.14 mm, respectively, in an investigation using lumbar MRI on 34 persons in 2016 [25]. Hong et al. reported anterior, middle and posterior disc heights of 10.83 mm, 10.05 mm and 7.20 mm, respectively, at the level of the L4–5 intervertebral disc in research conducted in the Korean population with lumbar MRI of 178 patients with a mean age of 15–25 years [19]. Kızılgöz and Ulusoy found that the anterior, middle and posterior disc heights at the level of L4–5 intervertebral discs were 13.26 mm, 11.20 mm and 8.44 mm, respectively, in their measurements performed with lateral lumbar spine radiography on 73 patients in 2019 [26]. In the MRI study of Lee et al. in 2017, in a Korean population of 389 people aged between 20 and 25 years, anterior and middle heights at L4–5 intervertebral disc levels were statistically significantly lower in both degenerated and herniated discs compared to normal intervertebral discs at this level [27]. In their study using lumbar MRI, Kızılgöz and Uzuner found that the intervertebral disc heights at the L4–5 level at the age of over 40 were statistically significantly lower in the group with disc pathology in the anterior, middle, and posterior heights compared to the normal group [4]. In our current study, the intervertebral disc height values at the L4–5 level were 12.34 mm, 11.58 mm and 7.60 mm in the anterior, middle and posterior, respectively. The measurement modality or genetic variants in distinct populations were assumed to be the causes of these similarities and differences.

In the current study, the age of the hernia group was found to be significantly higher than the normal group. This study also revealed that the anterior, middle, and posterior intervertebral disc heights at the L4–5 level of the normal group were found to be higher than the herniated group. Similar to our study, Kızılgöz and Uzuner’s study found that anterior, middle and posterior disc heights were statistically significantly lower in the group with herniated at L-4-5 intervertebral disc level [4].

In the study by Lee et al., the anterior and middle disc heights were found to be statistically significantly lower in the group with herniation at the level of the L-4-5 intervertebral disc. Our study found somewhat similar results with the study by Lee et al. [27].

This research consisted of one of the largest adult study populations to indicate the difference between herniated and non-herniated patient groups regarding intervertebral disc height and qualitative assessment of the discs using MRI. However, the results of this investigation should be examined carefully, considering the limitations of the study. First, herniated, and non-herniated groups were classified based on MRI results. Grouping patients regarding the surgical outcomes using the operated patients would be a lot better to strengthen the results. Factors such as clinical symptoms, smoking history and body mass index could not be evaluated in our study. The mean age differences between the study and control groups can be accepted as another limitation to discuss. Moreover, the reliability of the measurements is another aspect of this research. Even though the evaluations and measurements were carried out with two radiologists at the same time, the human error factor is still a matter to discuss. On the other hand, radiological assessments by the different radiologists using different monitors and calculating each measurement result to reveal interobserver agreements would give additional information to our colleagues.

## 5. Conclusions

L4–5 intervertebral disc heights were found to be lower in the herniated group compared to the control group, according to the results of this study. However, no significant difference was observed in terms of sex differences. Higher disc degeneration grades were found in the herniated group than in the normal group using Pfirrmann’s classification.

### Main Points

Higher Pfirmann grades were observed in herniated discsThe possibility of disc herniation increases with ageNo difference was observed between genders regarding the risk of disc herniation.

## Figures and Tables

**Figure 1 diagnostics-13-03190-f001:**
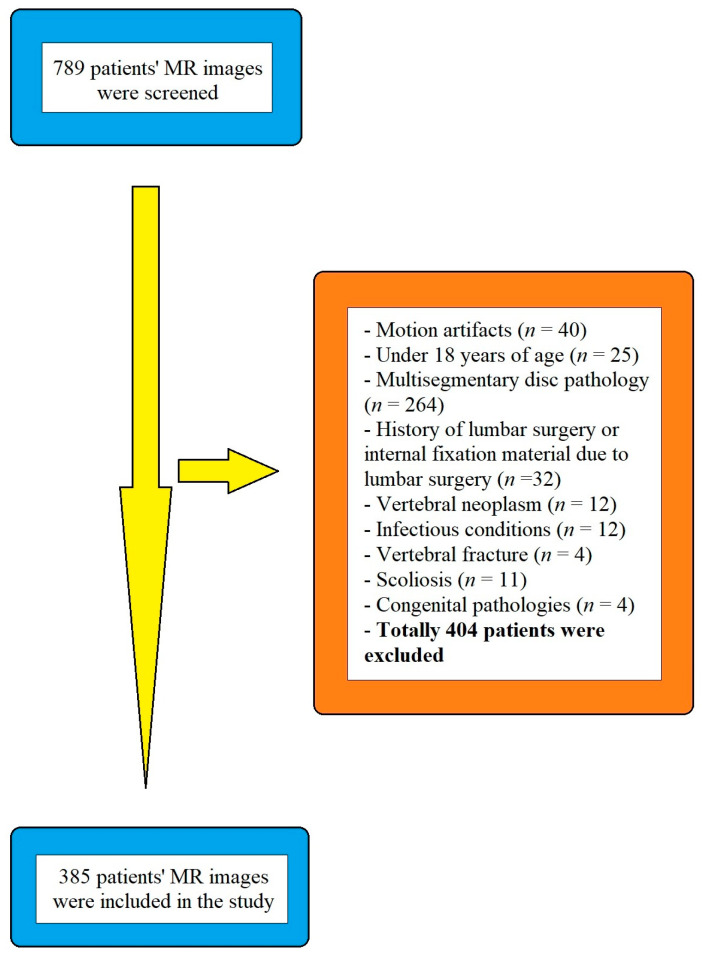
Diagram showing the study population.

**Figure 2 diagnostics-13-03190-f002:**
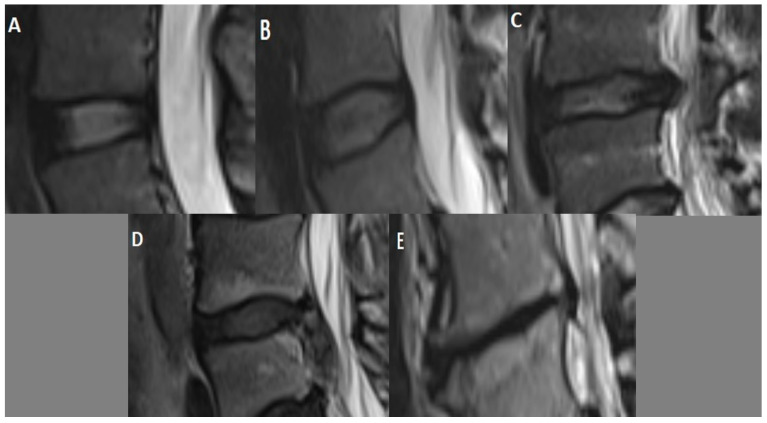
Pfirrmann grading scale for intervertebral disc degeneration with sagittal T2-weighted MRI sequence: Grade I (**A**)—homogeneous white normal disc; Grade II (**B**)—inhomogeneous white disc, normal disc height, may have horizontal bands; Grade III (**C**)—inhomogeneous gray disc, slightly reduced disc height; Grade IV (**D**)—inhomogeneous black disc with no clear distinction between the annulus and nucleus, marked loss of height in the disc; Grade V (**E**)—inhomogeneous black collapsed disc.

**Figure 3 diagnostics-13-03190-f003:**
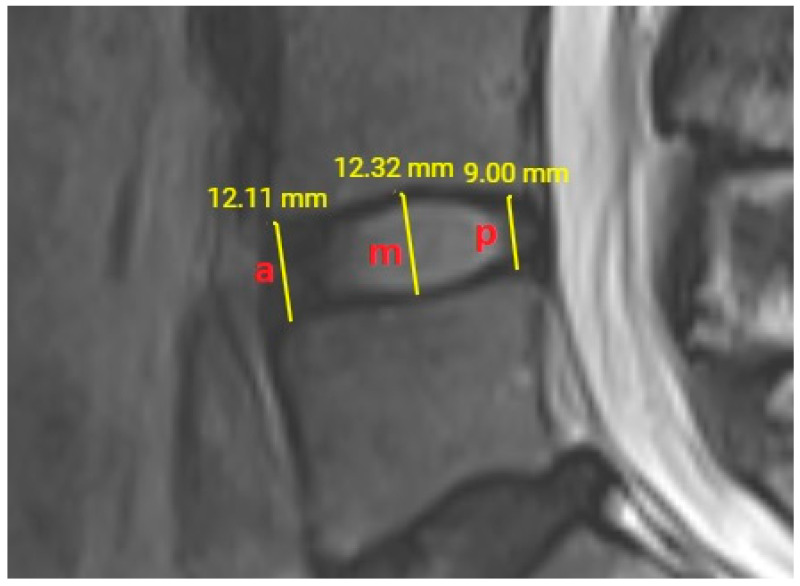
T2-weighted lumbar MRI shows anterior (a), middle (m) and posterior (p) measurements of height at L4–L5 intervertebral disc level.

**Table 1 diagnostics-13-03190-t001:** The sex distribution of the study population is shown.

Sex					
		Frequency	Percent	Valid Percent	Cumulative Percent
Valid	F	219	56.9	56.9	56.9
	M	166	43.1	43.1	100.0
	Total	385	100.0	100.0	

**Table 2 diagnostics-13-03190-t002:** The mean age of the groups in the study population is shown.

Group					
		Age Mean	Percent	Valid Percent	Cumulative Percent
Valid	Normal	44.73	74.0	74.0	74.0
	Hernie	52.25	26.0	26.0	100.0
	Total	46.68	100.0	100.0	

**Table 3 diagnostics-13-03190-t003:** Intervertebral disc height values at L4–5 level are shown for all patients in the study group.

		Int Disc Interval Anterior (mm)	Int Disc Interval Middle (mm)	Int Disc Interval Posterior (mm)	Int Disc Interval Mean (mm)
N	Valid	385	385	385	385
Mean		12.34	11.58	7.60	10.60
Median		12.48	11.92	7.63	10.66
Std. Deviation		2.36	2.38	1.53	2.50
Minimum		0.03	4.33	2.52	4.51
Maximum		18.78	17.74	11.66	44.70

**Table 4 diagnostics-13-03190-t004:** Intervertebral disc height values at L4–5 level are shown in the control group.

		Int Disc Interval Anterior (mm)	Int Disc Interval Middle (mm)	Int Disc Interval Posterior (mm)	Int Disc Interval Mean (mm)
N	Valid	285	285	285	285
Mean		12.57	11.92	7.79	10.88
Median		12.63	12.24	7.81	10.95
Std. Deviation		2.23	2.23	1.47	2.61
Minimum		0.03	4,33	2.52	4.91
Maximum		18.78	17.74	11.66	44.70

**Table 5 diagnostics-13-03190-t005:** Intervertebral disc height values at L4–5 level are shown in the hernia group.

		Int Disc Interval Anterior (mm)	Int Disc Interval Middle (mm)	Int Disc Interval Posterior (mm)	Int Disc Interval Mean (mm)
N	Valid	100	100	100	100
Mean		11.69	10.61	7.07	9.79
Median		11.93	10.68	7.00	10.08
Std. Deviation		2.61	2.53	1.58	1.96
Minimum		3.72	4.34	2.92	4.51
Maximum		17.23	16.49	11.25	13.06

**Table 6 diagnostics-13-03190-t006:** Distribution of L4–5 intervertebral disc level in the study population according to Pfirrmann’s grading system is shown.

Pfirrmann					
		Frequency	Percent	Valid Percent	Cumulative Percent
Valid	1	32	8.3	8.3	8.3
	2	78	20.3	20.3	28.6
	3	102	26.5	26.5	55.1
	4	140	36.4	36.4	91.4
	5	33	8.6	8.6	100.0
	Total	385	100.0	100.0	

**Table 7 diagnostics-13-03190-t007:** Distribution of L4–5 intervertebral disc level in the control group according to Pfirrmann’s grading system is shown.

Pfirrmann					
		Frequency	Percent	Valid Percent	Cumulative Percent
Valid	1	32	11.2	11.2	11.2
	2	75	26.3	26.3	37.5
	3	87	30.5	30.5	68.1
	4	78	27.4	27.4	95.4
	5	13	4.6	4.6	100.0
	Total	285	100.0	100.0	

**Table 8 diagnostics-13-03190-t008:** The distribution of the hernia group at the level of L4–5 intervertebral disc according to Pfirrmann’s grading system is shown.

Pfirrmann					
		Frequency	Percent	Valid Percent	Cumulative Percent
Valid	2	3	3.0	3.0	3.0
	3	15	15.0	15.0	18.0
	4	62	62.0	62.0	80.0
	5	20	20.0	20.0	100.0
	Total	100	100.0	100.0	

## Data Availability

The datasets generated and/or analyzed during the current study are not publicly available due to the risk of breach of patient data privacy but are available from the corresponding author on reasonable request.

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
