# Peer review of "The Comparison between Herniated and Non-Herniated Disc Levels Regarding Intervertebral Disc Space Height and Disc Degeneration, A Magnetic Resonance Study"

_diagnostics, 2023, doi:10.3390/diagnostics13203190_

Round 1

Reviewer 1 Report

I am a spine surgeon. Personally, I would not diagnose the presence of a herniated disc even if there is lumbar disc degeneration unless it is consistent with the physical findings of a herniated disc. (Especially if the herniation is at the foramen intervertebralis or outside the foramen intervertebralis, it may not be possible to make a determination based on MRI findings alone.)

As the authors note in their limitation, it is more plausible to validate the diagnosis using a group of patients who have undergone surgery for herniation.

However, we believe that the methodology of this study, which included a large number of patients and distinguished between herniated and normal groups with clear criteria regarding MRI images, is also scientific.

Please allow me to confirm the purpose of this study.

The purpose of this study is to determine the differences in disc height and levels of disc degeneration between herniated and non-herniated patients." It is obvious that disc herniation is caused by degeneration and that the intervertebral disc height decreases as the disc degenerates.

What you originally wanted to confirm was to examine the relationship between the qualitative evaluation using the Pfirrmann grading system and the qualitative measurement of intervertebral disc height, wasn't it?

Let me review the objectives of this study.

The purpose of this study is to determine the differences in disc height and level of disc degeneration between herniated and non-herniated patients.

However, it seems obvious to me that herniated discs are caused by degeneration and that as the disc degenerates the disc height decreases.

I believe what you originally wanted to confirm was to verify the relationship between qualitative and scientific changes. If so, I think we need to consider the relationship between Pfirrmann grade and intervertebral height.

If not, are you saying that you dared to quantitatively and scientifically confirm a clinically obvious event? (In Bone Joint J. 2013 Aug;95-B(8):1127-33. doi: 10.1302/0301-620X.95B8.31660. it is indeed stated that degenerative changes are not always observed before disc herniation occurs, and it may be a topic with little previous research I feel that this may be a topic that has not been studied much previously.)

If so, I would like you to reiterate the novelty unique to this study in light of previous studies by Kızılgöz and Uzuner et al. and Lee et al. Alternatively, please indicate the hypothesis you had before validation.

Author Response

Dear Editor and Reviewer of Diagnostics,

Thank you for your evaluation and kind reply. We appreciate your efforts and contributions to improving our manuscript (entitled " The comparison between herniated and non-herniated disc levels regarding intervertebral disc space height and disc degeneration; A magnetic resonance study” – Manuscript ID: diagnostics-2594396) and made the required corrections upon your recommendations.

Here are point-by-point answers to reviewers’ comments:

  1. Reviewer: I am a spine surgeon. Personally, I would not diagnose the presence of a herniated disc even if there is lumbar disc degeneration unless it is consistent with the physical findings of a herniated disc. (Especially if the herniation is at the foramen intervertebralis or outside the foramen intervertebralis, it may not be possible to make a determination based on MRI findings alone.). As the authors note in their limitation, it is more plausible to validate the diagnosis using a group of patients who have undergone surgery for herniation. However, I believe that the methodology of this study, which included a large number of patients and distinguished between herniated and normal groups with clear criteria regarding MRI images, is also scientific.

Answer
: We totally agree with these comments. The results of the operated patients would be more accurate. Instead of using the patients who had undergone surgery, we performed quantitative measurements and qualitative disc assessments meticulously to compare the herniated and non-herniated patients regarding appropriate statistical analyses. Many thanks to the Reviewer for these sincere and positive comments.   

  1. Reviewer: Please allow me to confirm the purpose of this study. The purpose of this study is to determine the differences in disc height and levels of disc degeneration between herniated and non-herniated patients. “It is obvious that disc herniation is caused by degeneration and that the intervertebral disc height decreases as the disc degenerates. What you originally wanted to confirm was to examine the relationship between the qualitative evaluation using the Pfirrmann grading system and the qualitative measurement of intervertebral disc height, wasn’t it. Let me review the objectives of this study. The purpose of this study is to determine the differences in disc height and level of disc degeneration between herniated and non-herniated patients. However, it seems obvious to me that herniated discs are caused by degeneration and that as the disc degenerates the disc height decreases. I believe what you originally confirm was to verify the relationship between qualitative and scientific changes. If so, I think we need to consider the relationship between Pfirrmann grade and intervertebral height.

If not, are you saying that you dared to quantitatively and scientifically confirm a clinically obvious event? (In Bone Joint J.2013 Aug;95-B(8):1127-33. doi: 10.1302/0301-620X.95B8.31660. It is needed stated that degenerative changes are not always observed before disc herniation occurs, and it may be a topic with previous research I feel that this may be a topic has not been studied much previously. If so, I would like you to reiterate the novelty unique to this study in light of previous studies by Kızılgöz and Uzuner et al. Alternatively, please indicate the hypothesis you had before validation.

Answer: The purpose of this study was to reveal the intervertebral disc space height and disc degeneration level differences between herniated and non-herniated patients.

The intervertebral disc space height and the presence of herniation was studied and herniated patients’ intervertebral disc height was lower than that of the non-herniated patients. 

The disc degeneration of herniated and non-herniated patients were compared and herniated patients’ disc degeneration level was found to be higher than that of the non-herniated patients.

Both results confirmed the expected situation regarding the degeneration process but indicates the result of a scientific analysis as a proof. In addition to these, the intervertebral disc space height measurements were performed at three locations as anterior, middle, and posterior height measurements. Besides, mean height was also used in statistical calculations. All these measurement results were lower for the herniated group, in comparison with the non-herniated patients. The results of the intervertebral disc space height measurements regarding these different locations were also presented with the other results.

The last paragraph of the introduction section was re-organized to explain the aim of the researchers of this study in detail.    

We are going to upload a “highlighted” version (to present the changes and corrections), a “clean” version of the revised manuscript to the submission system. We will be pleased to revise the manuscript if further revisions or corrections are requested by the Reviewer. Thank you for the opportunity to answer Reviewer's comments and your interest in our manuscript.

Reviewer 2 Report

Major comment that needs to be addressed in a revised manuscript.

1.       Unfortunately, the purpose of this study is unclear. What did the authors intend to show by evaluating the intervertebral disc height and intervertebral disc degeneration between the normal group and the group with disc herniation? Please clarify.

Specific comments

Introduction

2.       It would be desirable to include an additional explanation of why the authors chose the L4-5 level in this study in “Introduction'' rather than in “Discussion'' as is currently the case.

3.       In the second paragraph, “T2 sequences” should be T2-weighted sequences”.

Discussion

4.       It is desirable to clarify what this study has shown that is new and not found in previous reports.

Almost adequate.

Author Response

Dear Editor and Reviewer of Diagnostics,

Thank you for your evaluation and kind reply. We appreciate your efforts and contributions to improving our manuscript (entitled " The comparison between herniated and non-herniated disc levels regarding intervertebral disc space height and disc degeneration; A magnetic resonance study” – Manuscript ID: diagnostics-2594396) and made the required corrections upon your recommendations.

Here are point-by-point answers to reviewers’ comments:

  1. Reviewer: Major comment that needs to be addressed in a revised manuscript. 1.Unfortunately, the purpose of this study is unclear. What did the authors intend to show by evaluating the intervertebral disc height and intervertebral disc degeneration between the normal group and the group with disc herniation? Please clarify.

Answer
: In the last paragraph of the introduction section, new explanations were added to the text and highlighted in yellow as the Reviewer requested.

  1. Reviewer: Specific comments - Introduction

2.It would be desirable to include an additional explanation of why the authors choose the L4-5 level in this study in “introduction” rather than in “discussion” as is currently the case

Answer: The reason of choosing the L4-5 intervertebral disc level were explained in the introduction part and the additional sentence was highlighted in green.

  1. Reviewer: Specific comments – Introduction

3.In the second paragraph, “T2 sequences” should be “T2-weighted sequences”.

Answer: The required expression has now been corrected in the text as Reviewer requested (highlighted in green).

  1. Reviewer: Specific comments – Discussion

4.It is desirable to clarify what this study has shown that is new and not found in previous reports.

Answer: A new explanation was added as the first sentence of the limitations part of the discussion section as the Reviewer requested.

We are going to upload a “highlighted” version (to present the changes and corrections), a “clean” version of the revised manuscript to the submission system. We will be pleased to revise the manuscript if further revisions or corrections are requested by the Reviewer. Thank you for the opportunity to answer Reviewer's comments and your interest in our manuscript.